# Optimism in Reinforcement Learning with Generalized Linear Function Approximation

**Yining Wang**
University of Florida
yining.wang@warrington.ufl.edu

**Ruosong Wang**
Carnegie Mellon University
ruosongw@andrew.cmu.edu

**Simon S. Du**
University of Washington
ssdu@cs.washington.edu

**Akshay Krishnamurthy**
Microsoft Research
akshaykr@microsoft.com

## Abstract

We design a new provably efficient algorithm for episodic reinforcement learning with generalized linear function approximation. We analyze the algorithm under a new expressivity assumption that we call "optimistic closure," which is strictly weaker than assumptions from prior analyses for the linear setting. With optimistic closure, we prove that our algorithm enjoys a regret bound of $\widetilde{\mathcal{O}}\left(H\sqrt{d^3 T}\right)$ where $H$ is the horizon, $d$ is the dimensionality of the state-action features and $T$ is the number of episodes. This is the first statistically and computationally efficient algorithm for reinforcement learning with generalized linear functions.

## 1 Introduction

We study episodic reinforcement learning problems with infinitely large state spaces, where the agent must use function approximation to generalize across states while simultaneously engaging in strategic exploration. Such problems form the core of modern empirical/deep-RL, but relatively little work focuses on exploration, and even fewer algorithms enjoy strong sample efficiency guarantees.

On the theoretical side, classical sample efficiency results from the early 00s focus on "tabular" environments with small finite state spaces (Kearns & Singh, 2002; Brafman & Tennenholtz, 2002; Strehl et al., 2006), but as these methods scale with the number of states, they do not address problems with infinite or large state spaces. While this classical work has inspired practically effective approaches for large state spaces (Bellemare et al., 2016; Osband et al., 2016; Tang et al., 2017), these methods do not enjoy sample efficiency guarantees. More recent theoretical progress has produced provably sample efficient algorithms for complex environments where function approximation is required, but these algorithms are relatively impractical (Krishnamurthy et al., 2016; Jiang et al., 2017). In particular, these methods are computationally inefficient or rely crucially on strong dynamics assumptions (Du et al., 2019b).

In this paper, with an eye toward practicality, we study a simple variation of Q-learning, where we approximate the optimal Q-function with a generalized linear model. The algorithm is appealingly simple: collect a trajectory by following the greedy policy corresponding to the current model, perform a dynamic programming back-up to update the model, and repeat. The key difference over traditional Q-learning-like algorithms is in the dynamic programming step. Here we ensure that the updated model is *optimistic* in the sense that it always overestimates the optimal Q-function. This optimism is essential for our guarantees.

Optimism in the face of uncertainty is a well-understood and powerful algorithmic principle in short-horizon (e.g,. bandit) problems, as well as in tabular reinforcement learning (Azar et al., 2017; Dann et al., 2017; Jin et al., 2018). With linear function approximation, Yang & Wang (2019) and Jin et al. (2019) show that the optimism principle can also yield provably sample-efficient algorithms, when the environment dynamics satisfy certain linearity properties. Their assumptions are always satisfied in tabular problems, but are somewhat unnatural in settings where function approximation

is required. Moreover as these assumptions are directly on the dynamics, it is unclear how their analysis can accommodate other forms of function approximation, including generalized linear models.

In the present paper, we replace explicit dynamics assumptions with expressivity assumptions on the function approximator, and, by analyzing a similar algorithm to Jin et al. (2019), we show that the optimism principle succeeds under these strictly weaker assumptions.[1] More importantly, the relaxed assumption facilitates moving beyond linear models, and we demonstrate this by providing the first practical and provably efficient RL algorithm with generalized linear function approximation.

The paper is organized as follows: In Section 2 we formalize our setting, introduce the *optimistic closure* assumption, and discuss related assumptions in the literature. In Section 3 we study optimistic closure in detail and verify that it is strictly weaker than the recently proposed Linear MDP assumption. Our main algorithm and results are presented in Section 4, with the main proof in Section A. We close with some final remarks and future directions in Section 5.

## 2 PRELIMINARIES

We consider episodic reinforcement learning in a finite-horizon markov decision process (MDP) with possibly infinitely large state space $\mathcal{S}$, finite action space $\mathcal{A}$, initial distribution $\mu \in \Delta(\mathcal{S})$, transition operator $P : \mathcal{S} \times \mathcal{A} \to \Delta(\mathcal{S})$, reward function $R : \mathcal{S} \times \mathcal{A} \to \Delta([0,1])$ and horizon $H$. The agent interacts with the MDP in episodes and, in each episode, a trajectory $(s_1, a_1, r_1, s_2, a_2, r_2, \ldots, s_H, a_H, r_H)$ is generated where $s_1 \sim \mu$, for $h > 1$ we have $s_h \sim P(\cdot \mid s_{h-1}, a_{h-1})$, $r_h \sim R(s_h, a_h)$, and actions $a_{1:H}$ are chosen by the agent. For normalization, we assume that $\sum_{h=1}^{H} r_h \in [0,1]$ almost surely.

A (deterministic, nonstationary) policy $\pi = (\pi_1, \cdots, \pi_H)$ consists of $H$ mappings $\pi_h : \mathcal{S} \to \mathcal{A}$, where $\pi_h(s_h)$ denotes the action to be taken at time point $h$ if at state $s_h \in \mathcal{S}$ The *value* function for a policy $\pi$ is a collection of functions $(V_1^\pi, \ldots, V_H^\pi)$ where $V_h^\pi : \mathcal{S} \to \mathbb{R}$ is the expected future reward the policy collects if it starts in a particular state at time point $h$. Formally,

$$V_h^\pi(s) \triangleq \mathbb{E}\left[\sum_{h'=h}^{H} r_{h'} \mid s_h = s, a_{h:H} \sim \pi\right].$$

The value for a policy $\pi$ is simply $V^\pi \triangleq \mathbb{E}_{s_1 \sim \mu}[V_1^\pi(s_1)]$, and the optimal value is $V^\star \triangleq \max_\pi V^\pi$, where the maximization is over all nonstationary policies. The typical goal is to find an approximately optimal policy, and in this paper, we measure performance by the regret accumulated over $T$ episodes,

$$\text{Reg}(T) \triangleq TV^\star - \mathbb{E}\left[\sum_{t=1}^{T}\sum_{h=1}^{H} r_{h,t}\right].$$

Here $r_{h,t}$ is the reward collected by the agent at time point $h$ in the $t^{\text{th}}$ episode. We seek algorithms with regret that is sublinear in $T$, which demonstrates the agent's ability to act near-optimally over the long run.

### 2.1 Q-VALUES AND FUNCTION APPROXIMATION

For any policy $\pi$, the state-action value function, or the $Q$-function is a sequence of mappings $Q^\pi = (Q_1^\pi, \ldots, Q_H^\pi)$ where $Q_h^\pi : \mathcal{S} \times \mathcal{A} \to \mathbb{R}$ is defined as

$$Q_h^\pi(s, a) \triangleq \mathbb{E}\left[\sum_{h'=h}^{H} r_{h'} \mid s_h = s, a_h = a, a_{h+1:H} \sim \pi\right].$$

The optimal $Q$-function is $Q_h^\star \triangleq Q_h^{\pi^\star}$ where $\pi^\star \triangleq \text{argmax}_\pi V^\pi$ is the optimal policy.

In the value-based function approximation setting, we use a function class $\mathcal{G}$ to model $Q^\star$. In this paper, we always take $\mathcal{G}$ to be a class of generalized linear models (GLMs), defined as follows: Let $d \in \mathbb{N}$ be a dimensionality parameter and let $\mathbb{B}_d \triangleq \{x \in \mathbb{R}^d : \|x\|_2 \leq 1\}$ be the $\ell_2$ ball in $\mathbb{R}^d$.

---

[1]This is also mentioned as a remark in Jin et al. (2019).

**Definition 1.** *For a* known *feature map* $\phi : \mathcal{S} \times \mathcal{A} \to \mathbb{B}_d$ *and a* known *link function* $f : [-1, 1] \mapsto [-1, 1]$ *the class of* generalized linear models *is* $\mathcal{G} \triangleq \{(s, a) \mapsto f(\langle \phi(s, a), \theta \rangle) : \theta \in \mathbb{B}_d\}$.

As is standard in the literature (Filippi et al., 2010; Li et al., 2017), we assume the link function satisfies certain regularity conditions.

**Assumption 1.** $f(\cdot)$ *is either monotonically increasing or decreasing. Furthermore, there exist absolute constants* $0 < \kappa < K < \infty$ *and* $M < \infty$ *such that* $\kappa \le |f'(z)| \le K$ *and* $|f''(z)| \le M$ *for all* $|z| \le 1$.

For intuition, two example link functions are the identity map $f(z) = z$ and the logistic map $f(z) = 1/(1 + e^{-z})$ with bounded $z$. It is easy to verify that both of these maps satisfy Assumption 1.

## 2.2 EXPRESSIVITY ASSUMPTIONS: REALIZABILITY AND OPTIMISTIC CLOSURE

To obtain sample complexity guarantees that scale polynomially with problem parameters in the function approximation setting, it is necessary to posit expressivity assumptions on the function class $\mathcal{G}$ (Krishnamurthy et al., 2016; Du et al., 2019a). The weakest such condition is *realizability*, which posits that the optimal $Q$ function is in $\mathcal{G}$, or at least well-approximated by $\mathcal{G}$. Realizability alone suffices for provably efficient algorithms in the "contextual bandits" setting where $H = 1$ (Li et al., 2017; Filippi et al., 2010; Abbasi-Yadkori et al., 2011), but it does not seem to be sufficient when $H > 1$. Indeed in these settings it is common to make stronger expressivity assumptions (Chen & Jiang, 2019; Yang & Wang, 2019; Jin et al., 2019).

Following these works, our main assumption is a closure property of the *Bellman update* operator $\mathcal{T}_h$. This operator has type $\mathcal{T}_h : (\mathcal{S} \times \mathcal{A} \to \mathbb{R}) \to (\mathcal{S} \times \mathcal{A} \to \mathbb{R})$ and is defined for all $s \in \mathcal{S}, a \in \mathcal{A}$ as

$$\mathcal{T}_h(Q)(s, a) \triangleq \mathbb{E}\left[r_h + V_Q(s_{h+1}) \mid s_h = s, a_h = a\right],$$
$$V_Q(s) \triangleq \max_{a \in \mathcal{A}} Q(s, a).$$

The Bellman update operator for time point $H$ is simply $\mathcal{T}_H(Q)(s, a) \triangleq \mathbb{E}\left[r_H \mid s_H = s, a_H = a\right]$, which is degenerate. To state the assumption, we must first define the enlarged function class $\mathcal{G}_{\text{up}}$. For a $d \times d$ matrix $A$, $A \succeq 0$ denotes that $A$ is positive semi-definite. For a positive semi-definite matrix $A$, $\|A\|_{\text{op}}$ is the matrix operator norm, which is just the largest eigenvalue, and $\|x\|_A \triangleq \sqrt{x^\top A x}$ is the matrix Mahalanobis seminorm. For a fixed constant $\Gamma \in \mathbb{R}_+$ that we will set to be polynomial in $d$ and $\log(T)$, define

$$\mathcal{G}_{\text{up}} \triangleq \left\{(s, a) \mapsto 1 \wedge f(\langle \phi(s, a), \theta \rangle) + \gamma \|\phi(s, a)\|_A : \theta \in \mathbb{B}_d, A \succeq 0, \|A\|_{\text{op}} \le 1\right\},$$

Here we use $a \wedge b \triangleq \min\{a, b\}$. The class $\mathcal{G}_{\text{up}}$ contains $\mathcal{G}$ in addition to all possible upper confidence bounds that arise from solving least squares regression problems using the class $\mathcal{G}$. We now state our main expressivity assumption, which we call *optimistic closure*.

**Assumption 2** (Optimistic closure). *For any* $1 \le h < H$ *and* $g \in \mathcal{G}_{up}$, *we have* $\mathcal{T}_h(g) \in \mathcal{G}$.

In words, when we perform a Bellman backup on any upper confidence bound function for time point $h + 1$, we obtain a generalized linear function at time $h$. While this property seems quite strong, we note that a similar notion is mentioned informally in Jin et al. (2019) and that related closure-type assumptions are common in the literature (see Section 2.3 for detailed discussion). More importantly, we will prove in Section 3 that optimistic closure is actually *strictly weaker* than previous assumptions used in our RL setting where exploration is required. Before turning to these discussions, we mention two basic properties of optimistic closure.

**Fact 1** (Optimistic closure and realizability). *Optimistic closure implies that* $Q^\star \in \mathcal{G}$ *(realizability)*.

*Proof.* We will solve for $Q^\star$ via dynamic programming, starting from time point $H$. In this case, the Bellman update operator is degenerate, and we start by observing that $\mathcal{T}_H(g) \equiv Q_H^\star$ for all $g$. Consequently we have $Q_H^\star \in \mathcal{G}$. Next, inductively we assume that we have $Q_{h+1}^\star \in \mathcal{G}$, which implies that $Q_{h+1}^\star \in \mathcal{G}_{\text{up}}$ as we may take the same parameter $\theta$ and set $A \equiv 0$. Then, by the standard Bellman fixed-point characterization, we know that $Q_h^\star = \mathcal{T}_h(Q_{h+1}^\star)$, at which point Assumption 2 yields that $Q_h^\star \in \mathcal{G}$. $\qquad \square$

**Fact 2** (Optimistic closure in tabular settings). *If $\mathcal{S}$ is finite and $\phi(s,a) = e_{s,a}$ is the standard-basis feature map, then under Assumption 1 we have optimistic closure.*

*Proof.* We simply verify that $\mathcal{G}$ contains *all* mappings from $(s,a) \mapsto [0,1]$, at which point the result is immediate. To see why, observe that via Assumption 1 we know that $f$ is invertible (it is monotonic with derivative bounded from above and below). Then, note that any function $(s,a) \mapsto [0,1]$ can be written as a vector $v \in [0,1]^{|\mathcal{S}| \times |\mathcal{A}|}$. For such a vector $v$, if we define $\theta_{s,a} \triangleq f^{-1}(v_{s,a})$ we have that $f(\langle e_{s,a}, \theta \rangle) = v_{s,a}$. Hence $\mathcal{G}$ contains all functions, so we trivially have optimistic closure. $\square$

## 2.3 RELATED WORK

The majority of the theoretical results for reinforcement learning focus on the *tabular* setting where the state space is finite and sample complexities scaling polynomially with $|\mathcal{S}|$ are tolerable (Kearns & Singh, 2002; Brafman & Tennenholtz, 2002; Strehl et al., 2006). Indeed, by now there are a number of algorithms that achieve strong guarantees in this setting (Dann et al., 2017; Azar et al., 2017; Jin et al., 2018; Simchowitz & Jamieson, 2019). Via Fact 2, our results apply to this setting, and indeed our algorithm can be viewed as a generalization of the canonical tabular algorithm (Azar et al., 2017; Dann et al., 2017; Simchowitz & Jamieson, 2019) to the function approximation setting.[2]

Turning to the function approximation setting, several other results concern function approximation in settings where exploration is not an issue, including the infinite-data regime (Munos, 2003; Farahmand et al., 2010) and the "batch RL" setting where the agent does not control the data-collection process (Munos & Szepesvári, 2008; Antos et al., 2008; Chen & Jiang, 2019). While the details differ, all of these results require that the function class satisfy some form of (approximate) closure with respect to the Bellman operator. As an example, one assumption is that $\mathcal{T}(g) \in \mathcal{G}$ for all $g \in \mathcal{G}$, with an appropriately defined approximate variant (Chen & Jiang, 2019). These results therefore provide motivation for our optimistic closure assumption. While optimistic closure is stronger than the assumptions in these works, we emphasize that we are also addressing exploration, so our setting is also significantly more challenging.

A recent line of work studies function approximation in settings where the agent must explore the environment (Krishnamurthy et al., 2016; Jiang et al., 2017; Du et al., 2019b). The algorithms developed here can accommodate function classes beyond generalized linear models, but they are still relatively impractical and the more practical ones require strong dynamics assumptions (Du et al., 2019b). In contrast, our algorithm is straightforward to implement and does not require any explicit dynamics assumption. As such, we view these results as complementary to our own.

Most closely related to our work are the recent results of Yang & Wang (2019) and Jin et al. (2019). Both papers study MDPs with certain linear dynamics assumptions (what they call the Linear MDP assumption) and use linear function approximation to obtain provably efficient algorithms. Jin et al. (2019) hint at optimistic closure as a weakening of their Linear MDP assumption and remark that their guarantees continues to hold under this weaker assumption. One of our contributions is to formalize this remark. Indeed, our algorithm is almost identical to theirs. However we emphasize that optimistic closure is strictly weaker than their Linear MDP assumption, which in turn is strictly weaker than the assumption of Yang & Wang (2019). Further, and perhaps more importantly, by avoiding explicit dynamics assumptions, we enable approximation with GLMs, which are incompatible with the Linear MDP structure. Hence, the present paper can be seen as a significant generalization of these recent results.

Since the initial version of this paper appeared, several other works have studied linear function approximation in reinforcement learning. A number of papers (Cai et al., 2019; Ayoub et al., 2020; Modi et al., 2020; Zhou et al., 2020) study an incomparable class of dynamics models that permit linear function approximation. Others study weakenings of the Linear MDP assumptions. In particular, Agarwal et al. (2020) only require small transfer error for linear regression, which formalizes out-of-distribution generalization and is always zero in Linear MDPs. Zanette et al. (2020a) only require that the Bellman operator is closed with respect to linear functions, which is considerably

---

[2]The description of the algorithm looks quite different from that of Azar et al. (2017), but via an equivalence between model-free methods with experience replay and model-based methods (Fujimoto et al., 2018), they are indeed quite similar.

weaker than our optimistic closure assumption. However, their algorithm is not computationally efficient. Computational efficiency is addressed in Zanette et al. (2020b) in the reward-free setting with reachability assumptions. As we do not require reachability assumptions, this latter result is incomparable to ours. None of these results considers generalized linear models.

## 3   ON OPTIMISTIC CLOSURE

For a more detailed comparison to the recent work of Yang & Wang (2019) and Jin et al. (2019), we define the linear MDP model studied in the latter work.

**Definition 2.** *An MDP is said to be a* linear MDP *if there exist known feature map* $\psi : \mathcal{S} \times \mathcal{A} \to \mathbb{R}^d$, *unknown signed measures* $\mu : \mathcal{S} \to \mathbb{R}^d$, *and an unknown vector* $\eta \in \mathbb{R}^d$ *such that (1)* $P(s'|s,a) = \langle \psi(s,a), \mu(s') \rangle$ *holds for all states* $s, s'$ *and actions* $a$, *and (2)* $\mathbb{E}[r \mid s,a] = \langle \psi(s,a), \eta \rangle$.

Linear MDPs are studied by Jin et al. (2019), who establish a $\sqrt{T}$-type regret bound for an optimistic algorithm. This assumption already subsumes that of Yang & Wang (2019), and related assumptions also appear elsewhere in the literature (Bradtke & Barto, 1996; Melo & Ribeiro, 2007; Zanette et al., 2019). In this section, we show that optimistic closure (Assumption 2) is strictly weaker than assuming the environment is a linear MDP.

**Proposition 1.** *If an MDP is linear then Assumption 2 holds with* $\mathcal{G} = \{(s,a) \mapsto \langle w, \psi(s,a) \rangle : w \in \mathbb{B}_d\}$.

*Proof.* The result is implicit in Jin et al. (2019), and we include the proof for completeness. For any function $g$, observe that owing to the linear MDP property

$$\mathcal{T}_h(g)(s,a) = \mathbb{E}\left[r + \max_{a'} g(s',a') \mid s,a\right] = \langle \psi(s,a), \eta \rangle + \int \langle \psi(s,a), \mu(s') \rangle \max_{a'} g(s',a') \mathrm{d}s',$$

which is clearly a linear function in $\psi(s,a)$. Hence for any function $g$, which trivially includes the optimistic functions, we have $\mathcal{T}_h(g) \in \mathcal{G}$. □

Thus the linear MDP assumption is stronger than Assumption 2. Next, we show that it is strictly stronger.

**Proposition 2.** *There exists an MDP with* $H = 2$, $d = 2$, $|\mathcal{A}| = 2$ *and* $|\mathcal{S}| = \infty$ *such that Assumption 2 is satisfied, but the MDP is not a linear MDP.*

Thus we have that optimistic closure is strictly weaker than the linear MDP assumption from Jin et al. (2019). Thus, our results strictly generalize theirs.

*Proof.* Fix the link function to be $f(z) = z$. We first construct the MDP. Set the action space $\mathcal{A} = \{a_1, a_2\}$. We use $e_i$ to denote the $i^{\text{th}}$ standard basis element, and let $x = (0.1/\Gamma, 0.1/\Gamma)$ be a fixed vector where $\Gamma$ appears in the construction of $\mathcal{G}_{\text{up}}$. Recall that $s_1$ is the first state in each trajectory. In our example, for all $a \in \mathcal{A}$, $\phi(s_1, a)$ is sampled uniformly at random from the set $\{\alpha e_1 + (1 - \alpha)e_2 : \alpha \in [0, 1]\}$. The transition rule is deterministic and given by:

$$\forall a \in \mathcal{A} : \phi(s_2, a) = \alpha x \text{ if } \phi(s_1, a) = \alpha e_1 + (1 - \alpha)e_2.$$

Moreover, for the reward function, $R(s_1, a) = 0$ and $R(s_2, a) = 0.1\alpha/\Gamma$.

We first show that the Linear MDP property does not hold for the constructed MDP and the given feature map $\phi$. Let $s_1^{(1)}$ be the state with $\phi(s_1^{(1)}, a) = e_2$ and $s_1^{(2)}$ be the state with $\phi(s_1^{(2)}, a) = e_1$. Notice that we deterministically transition from $s_1^{(1)}$ to a state $s_2^{(1)}$ with $\phi(s_2^{(1)}, a) = 0$, and we deterministically transition from $s_1^{(2)}$ to a state $s_2^{(2)}$ with $\phi(s_2^{(2)}, a) = x$, which already fixes the whole transition operator under the linear MDP assumption. Thus, under the linear MDP assumption, we must therefore have a randomized transition for any state $s_1$ with $\phi(s_1, a) = \alpha e_1 + (1 - \alpha)e_2$ where $\alpha \in (0, 1)$. This contradicts the fact that our constructed MDP has deterministic transitions everywhere, so the linear MDP cannot hold.

We next show that Assumption 2 holds. Consider an arbitrary optimistic $Q$ estimate of the form $g(z) = \min\{1, z^\top \theta + \gamma \sqrt{z^\top A z}\} \in \mathcal{G}_{\text{up}}$. Notice that for $x = (0.1/\Gamma, 0.1/\Gamma)$, we always have that

---

**Algorithm 1** The LSVI-UCB algorithm with generalized linear function approximation.

1: Initialize estimates $\bar{Q}_{h,0} \equiv 1$ for all $h \leq H$ and $\bar{Q}_{H+1,t} \equiv 0$ for all $1 \leq t \leq T$;
2: Set $\gamma = CK\kappa^{-1}\sqrt{1 + M + K + d^2 \ln((1 + K + \Gamma)TH)}$ for a universal constant $C$;
3: **for** $t = 1, 2, \cdots, T$ **do**
4:      Commit to policy $\hat{\pi}_{h,t}(s) \triangleq \operatorname{argmax}_{a \in \mathcal{A}} \bar{Q}_{h,t-1}(s, a)$;
5:      Use policy $\hat{\pi}_{\cdot,t}$ to collect one trajectory $\{(s_{h,t}, a_{h,t}, r_{h,t})\}_{h=1}^{H}$;
6:      **for** $h = H, H - 1, \cdots, 1$ **do**
7:          Compute $x_{h,\tau} \triangleq \phi(s_{h,\tau}, a_{h,\tau})$ and $y_{h,\tau} \triangleq r_{h,\tau} + \max_{a' \in \mathcal{A}} \bar{Q}_{h+1,t}(s_{h+1,\tau}, a')$ for all $\tau \leq t$;
8:          Compute ridge estimate

$$\hat{\theta}_{h,t} \triangleq \operatorname*{argmin}_{\|\theta\|_2 \leq 1} \sum_{\tau \leq t} (y_{h,\tau} - f(\langle x_{h,\tau}, \theta \rangle))^2; \tag{1}$$

9:          Compute $\Lambda_{h,t} \triangleq \sum_{\tau \leq t} x_{h,\tau} x_{h,\tau}^{\top} + I$;
10:         Construct $\bar{Q}_{h,t}(s, a) \triangleq \min\left\{1, f(\phi(s,a)^{\top}\hat{\theta}_{h,t}) + \gamma \|\phi(s,a)\|_{\Lambda_{h,t}^{-1}}\right\}$;
11:      **end for**
12: **end for**

---

$x^{\top}\theta + \gamma\sqrt{x^{\top}Ax} \leq 1$ for any $\theta \in \mathbb{B}_d$ and $A$ with $\|A\|_{\mathrm{op}} \leq 1$. Moreover, for all $s_2$, i.e., the second state in the trajectory, we always have $\phi(s_2, a) = \alpha x$ for some $\alpha \in [0, 1]$. Hence we can ignore the first term in the minimum, and, by direct calculation, we have that when $\phi(s, a) = \alpha e_1 + (1 - \alpha)e_2$:

$$\mathcal{T}_1(g)(s, a) = \alpha x^{\top}\theta + \gamma\sqrt{\alpha^2 x^{\top} Ax}$$
$$= \alpha(x^{\top}\theta + \gamma\sqrt{x^{\top}Ax}) = \alpha c_0.$$

Hence we can write $\mathcal{T}_1(g) = \langle \phi(s, a), (c_0, 0) \rangle$, which verifies Assumption 2. $\qquad\square$

## 4 ALGORITHM AND MAIN RESULT

We now turn to presenting our algorithm and main results. We study a least-squares dynamic programming style algorithm that we call LSVI-UCB, with pseudocode presented in Algorithm 1. The algorithm is nearly identical to the algorithm proposed by Jin et al. (2019) with the same name. A similar algorithmic template has also been extensively studied in the tabular setting Azar et al. (2017); Dann et al. (2017); Simchowitz & Jamieson (2019), albeit with slightly different confidence bounds. As our algorithm applies to all of these settings, it should be considered as a generalization.

The algorithm uses dynamic programming to maintain optimistic $Q$ function estimates $\{\bar{Q}_{h,t}\}_{h \leq H, t \leq T}$ for each time point $h$ and each episode $t$. In the $t^{\text{th}}$ episode, we use the previously computed estimates to define the greedy policy $\hat{\pi}_{h,t}(\cdot) \triangleq \operatorname{argmax}_{a \in \mathcal{A}} \bar{Q}_{h,t-1}(\cdot, a)$, which we use to take actions for the episode. Then, with all of the trajectories collected so far, we perform a dynamic programming update, where the main per-step optimization problem is (1). Starting from time point $H$, we update our $Q$ function estimates by solving constrained least squares problems using our class of GLMs. At time point $H$, the covariates are $\{\phi(s_{H,\tau}, a_{H,\tau})\}_{\tau \leq t}$, and the regression targets are simply the immediate rewards $\{r_{H,\tau}\}_{\tau \leq t}$. For time points $h < H$, the covariates are defined similarly as $\{\phi(s_{h,\tau}, a_{h,\tau})\}_{\tau \leq t}$ but the regression targets are defined by inflating the learned $Q$ function for time point $h + 1$ by an *optimism bonus*.

In detail, the least squares problem for time point $h + 1$ yields a parameter $\hat{\theta}_{h+1,t}$ and we also form the second moment matrix $\Lambda_{h+1,t}$ of all the covariates at time $h + 1$ that we have seen so far. Using these, we define the optimistic $Q$ function $\bar{Q}_{h+1,t}(s, a) \triangleq \min\left\{1, f(\langle \phi(s, a), \hat{\theta}_{h+1,t} \rangle) + \gamma \|\phi(s, a)\|_{\Lambda_{h+1,t}^{-1}}\right\}$. In our analysis, we verify that $\bar{Q}_{h+1,t}$ is optimistic in the sense that it over-estimates $Q^{\star}$ for every $(s, a)$. Then, the regression targets for the least squares problem at time point $h$ are $r_{h,\tau} + \max_{a' \in \mathcal{A}} \bar{Q}_{h+1,t}(s_{h+1,\tau}, a')$, which is a natural stochastic approximation to the Bellman backup of $\bar{Q}_{h+1,t}$. Applying this update backward from

time point $H$ to 1, we obtain the $Q$-function estimates that we use to define the policy for the next episode.

The main conceptual difference between Algorithm 1 and the algorithm of Jin et al. (2019) is that we allow non-linear function approximation with GLMs, while they consider only linear models. On a more technical level, we use constrained least squares for our dynamic programming backup which we find easier to analyze, while they use the ridge regularized version.

On the computational side, the algorithm is straightforward to implement, and, depending on the link function $f$, it can be easily shown to run in polynomial time. For example, if $f$ is the identity map, then (1) is equivalent to standard least square ridge regression, which can be solved in closed form. Moreover, we can use the Sherman-Morrison formula to amortize matrix inversions, and, by doing so, we obtain a running time of $O\left(d^2|\mathcal{A}|HT^2\right)$. The dominant cost in this calculation is evaluating the optimism bonus when computing the regression targets. In practice, using an epoch schedule or incremental optimization algorithms for updating $\bar{Q}$ would yield an even faster algorithm. Of course, with modern machine learning libraries, it is also straightforward to implement the algorithm with a non-trivial link function $f$, even though (1) may be non-convex.

## 4.1 MAIN RESULT

Our main result is a regret bound for LSVI-UCB when the link function satisfies Assumption 1 and the function class satisfies Assumption 2.

**Theorem 1.** *For any episodic MDP, with Assumption 1 and Assumption 2, and for any $T$, the cumulative regret of Algorithm 1 is*[3]

$$O\left(HK\kappa^{-1}\sqrt{(M+K+d^2\ln(KTH))\cdot Td\ln(T/d)}\right) = \widetilde{\mathcal{O}}\left(H\sqrt{d^3T}\right),$$

*with probability $1 - 1/(TH)$.*

The result states that LSVI-UCB enjoys $\sqrt{T}$-regret for any episodic MDP problem and any GLM, provided that the regularity conditions are satisfied and that optimistic closure holds. As we have mentioned, these assumptions are relatively mild, encompassing the tabular setting and prior work on linear function approximation. Importantly, no explicit dynamics assumptions are required. Thus, Theorem 1 is one of the most general results we are aware of for provably efficient exploration with function approximation.

Nevertheless, to develop further intuition for our bound, it is worth comparing to prior results. First, in the linear MDP setting of Jin et al. (2019), we use the identity link function so that $K = \kappa = 1$ and $M = 1$, and we also are guaranteed to satisfy Assumption 2. In this case, our bound differs from that of Jin et al. (2019) only in the dependence on $H$, which arises due to a difference in normalization. Our bound is essentially equivalent to theirs and can therefore be seen as a strict generalization.

To capture the tabular setting, we use the standard basis featurization as in Fact 2 and the identity link function, which gives $d = |\mathcal{S}||\mathcal{A}|$, $K = \kappa = 1$, and $M = 1$. Thus, we obtain the following corollary:

**Corollary 2.** *For MDPs with finite state and action spaces, using feature map $\phi(s,a) \triangleq e_{s,a} \in \mathbb{R}^{|\mathcal{S}|\times|\mathcal{A}|}$, for any $T$, the cumulative regret of Algorithm 1 is $\widetilde{\mathcal{O}}\left(H\sqrt{|\mathcal{S}|^3|\mathcal{A}|^3T}\right)$, with probability $1 - 1/(TH)$.*

Note that this bound is polynomially worse than the near-optimal $\widetilde{\mathcal{O}}(H\sqrt{SAT} + H^2S^2A\log(T))$ bound of Azar et al. (2017). However, a refined analysis specialized to the tabular setting can be shown to obtain a better regret bound of $\widetilde{\mathcal{O}}\left(H\sqrt{|\mathcal{S}|^2|\mathcal{A}|^2T}\right)$. Of course, our algorithm and analysis address problems with infinitely large state spaces and other settings that are significantly more complex than tabular MDPs, which we believe is more important than recovering the optimal guarantee for tabular MDPs.

---

[3] We use $\widetilde{\mathcal{O}}\left(\cdot\right)$ to suppress factors of $M, K, \kappa, \Gamma$ and any logarithmic dependencies on the arguments.

## 5 DISCUSSION

This paper presents a provably efficient reinforcement learning algorithm that approximates the $Q^\star$ function with a generalized linear model. We prove that the algorithm obtains $\widetilde{\mathcal{O}}(H\sqrt{d^3T})$ regret under mild regularity conditions and a new expressivity condition that we call *optimistic closure*. These assumptions generalize both the tabular setting, which is classical, and the linear MDP setting studied in recent work. Further they represent the first statistically and computationally efficient algorithms for reinforcement learning with generalized linear function approximation, without explicit dynamics assumptions.

We close with some open problems. First, using the fact that Corollary 3 applies beyond GLMs, can we develop algorithms that can employ general function classes? While such algorithms do exist for the contextual bandit setting (Foster et al., 2018), it seems quite difficult to generalize this analysis to multi-step reinforcement learning. More importantly, while optimistic closure is weaker than some prior assumptions (and incomparable to others), it is still quite strong, and stronger than what is required for the batch RL setting. An important direction is to investigate weaker assumptions that enable provably efficient reinforcement learning with function approximation. We look forward to studying these questions in future work.

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

## A  PROOF OF THEOREM 1

We now provide the proof of Theorem 1, deferring some technical details to later sections in this appendix. The proof has three main components: a regret decomposition for optimistic $Q$ learning, a deviation analysis for least squares with GLMs to ensure optimism, and a potential argument to obtain the final regret bound.

**Regret decomposition.**  The first step of the proof is a regret decomposition that applies generically to optimistic algorithms.[4] The lemma demonstrates concisely the value of optimism in reinforcement learning, and is the primary technical motivation for our interest in optimistic algorithms.

We state the lemma more generally, which requires some additional notation. Fix round $t$ and let $\{\bar{Q}_{h,t-1}\}_{h\leq H}$ denote the current estimated $Q$ functions. The precondition is that $\bar{Q}_{h,t-1}$ is optimistic and has controlled overestimation. Precisely, we assume that there exists a function $\mathrm{cnf}_{h,t-1}: \mathcal{S} \times \mathcal{A} \to \mathbb{R}_+$ such that

$$Q_h^\star(s,a) \leq \bar{Q}_{h,t-1}(s,a) \tag{2}$$

$$\bar{Q}_{h,t-1}(s,a) \leq \mathcal{T}_h(\bar{Q}_{h+1,t-1})(s,a) + \mathrm{cnf}_{h,t-1}(s,a) \tag{3}$$

We will verify that our estimates $\bar{Q}_{h,\cdot}$ satisfy these properties subsequently. Before doing so, we state the regret decomposition lemma and an immediate corollary.

**Lemma 1.** *Fix episode $t$ and let $\mathcal{F}_{t-1}$ be the filtration of $\{(s_{h,\tau}, a_{h,\tau}, r_{h,\tau})\}_{\tau<t}$. Assume that $\bar{Q}_{h,t-1}$ satisfies (3) for some function $\mathrm{cnf}_{h,t-1}$. Then, if $\pi_t = \mathrm{argmax}_{a\in\mathcal{A}}\bar{Q}_{h,t-1}(\cdot,a)$ is deployed we have*

$$V^\star - \mathbb{E}\left[\sum_{h=1}^H r_{h,t} \mid \mathcal{F}_{t-1}\right] \leq \zeta_t + \sum_{h=1}^H \mathrm{cnf}_{h,t-1}(s_{h,t}, a_{h,t}),$$

*where $\mathbb{E}[\zeta_t \mid \mathcal{F}_{t-1}] = 0$ and $|\zeta_t| \leq 2H$ almost surely.*

**Corollary 3.** *Assume that for all $t$, $\bar{Q}_{h,t-1}$ satisfies (3) and that $\pi_t$ is the greedy policy with respect to $\bar{Q}_{h,t-1}$. Then with probability at least $1 - \delta$, we have*

$$\mathrm{Reg}(T) \leq \sum_{t=1}^T \sum_{h=1}^H \mathrm{cnf}_{h,t-1}(s_{h,t}, a_{h,t}) + O(H\sqrt{T\log(1/\delta)}).$$

*Proof of Lemma 1.* Observe that

$$\begin{aligned}
V^\star = \mathbb{E}[Q^\star(s_1, \pi^\star(s_1))] &\leq \mathbb{E}[\bar{Q}_{1,t-1}(s_1, \pi^\star(s_1))] \\
&\leq \mathbb{E}[\bar{Q}_{1,t-1}(s_1, \pi_t(s_1))] \\
&\leq \mathbb{E}[\mathrm{cnf}_{1,t-1}(s_1, \pi_t(s_1))] + \mathbb{E}[\mathcal{T}_1(\bar{Q}_{2,t-1})(s_1, \pi_t(s_1))] \\
&= \mathbb{E}[\mathrm{cnf}_{1,t-1}(s_1, \pi_t(s_1))] + \mathbb{E}[r_1 \mid s_1, a_1 = \pi_t(s_1)] \\
&\quad + \mathbb{E}_{s_2\sim\pi_t}[\bar{Q}_{2,t-1}(s_2, \pi_t(s_2))]
\end{aligned}$$

---

[4]Related results appear elsewhere in the literature focusing on the tabular setting, see e.g., Simchowitz & Jamieson (2019).

Throughout this calculation, $s_1 \sim \mu$. The first step here is by definition, the second uses the optimism property for $\bar{Q}_{1,t-1}$. The third uses that $\pi_t$ is the greedy policy with respect to $\bar{Q}_{1,t-1}$ while the fourth uses the upper bound on $\bar{Q}_{1,t-1}$. Finally we use the definition of the Bellman operator and the fact that $\pi_t$ is the greedy policy yet again. Comparing this upper bound with the expected reward collected by $\pi_t$ we observe that $r_1$ cancels, and we get

$$V^\star - \mathbb{E}\left[\sum_{h=1}^H r_{h,t} \mid \mathcal{F}_{t-1}\right] \leq \mathbb{E}_{\pi_t}\left[\mathrm{cnf}_{1,t-1}(s_1, \pi_t(s_1))\right] + \mathbb{E}_{\pi_t}\left[\bar{Q}_{2,t-1}(s_2, \pi_t(s_2)) - \sum_{h=2}^H r_{h,t} \mid \mathcal{F}_{t-1}\right].$$

At this point, notice that $\bar{Q}_{2,t-1}(s_2, \pi_t(s_2))$ is precisely what we alreacy upper bounded at time point $h = 1$ and we are always considering the state-action distribution induced by $\pi_t$. Hence, repeating the argument for all $h$, we obtain

$$V^\star - \mathbb{E}\left[\sum_{h=1}^H r_{h,t} \mid \mathcal{F}_{t-1}\right] \leq \sum_{h=1}^H \mathbb{E}_{\pi_t}\left[\mathrm{cnf}_{h,t-1}(s_h, a_h)\right] = \sum_{h=1}^H \mathrm{cnf}_{h,t-1}(s_{h,t}, a_{h,t}) + \zeta_t,$$

where $\zeta_t \triangleq \sum_{h=1}^H \zeta_{h,t}$ and

$$\zeta_{h,t} \triangleq \mathbb{E}_{\pi_t}\left[\mathrm{cnf}_{h,t-1}(s_h, \pi_t(s_h))\right] - \mathrm{cnf}_{h,t-1}(s_{h,t}, a_{h,t}),$$

which is easily seen to have the required properties. □

The lemma states that if $\bar{Q}_{h,t-1}$ is optimistic and we deploy the greedy policy $\pi_t$, then the per-episode regret is controlled by the overestimation error of $\bar{Q}_{h,t-1}$, up to a stochastic term that enjoys favorable concentration properties. Crucially, the errors are accumulated on the observed trajectory, or, stated another way, the $\mathrm{cnf}_{h,t-1}$ is evaluated on the states and actions visited during the episode. As these states and actions will be used to update $\bar{Q}$, we can expect that the cnf function will decrease on these arguments. This can yield one of two outcomes: either we will incur lower regret in the next episode, or we will explore the environment by visiting new states and actions. In this sense, the lemma demonstrates how optimism navigates the exploration-exploitation tradeoff in the multi-step RL setting, analogously to the bandit setting.

Note that Lemma 1 does not assume any form for $\bar{Q}_{h,t-1}$ and does not require Assumption 2. In particular, they are not specialized to GLMs. In our proof, we use the GLM representation and Assumption 2 to ensure that (3) holds and to bound the confidence sum in Corollary 3. We believe these technical results will be useful in designing RL algorithms for general function classes, which is a natural direction for future work.

**Deviation analysis.** The next step of the proof is to design the cnf function and ensure that (3) holds, with high probability. This is the contents of the next lemma.

**Lemma 2.** *Under Assumption 1 and Assumption 2, with probability $1 - 1/(TH)$, we have that $\forall t, h, s, a$:*

$$\left|f(\langle \phi(s,a), \hat{\theta}_{h,t}\rangle) - \mathcal{T}_h(\bar{Q}_{h+1,t})(s,a)\right| \leq \gamma \|\phi(s,a)\|_{\Lambda_{h,t}^{-1}}$$

*where $\gamma, \Lambda_{h,t}$ are defined in Algorithm 1.*

A simple induction argument then verifies that (3) holds, which we summarize in the next corollary.

**Corollary 4.** *Under Assumption 1 and Assumption 2, with probability $1 - 1/(TH)$, we have that (3) holds for all $t, h$ with $\mathrm{cnf}_{h,t-1}(s,a) = \min\{2, 2\gamma \|\phi(s,a)\|_{\Lambda_{h,t-1}^{-1}}\}$.*

As the proof of Lemma 2 is rather long and technical, we defer the details to the appendix and instead explain the high-level argument here. The proof requires an intricate deviation analysis to account for the dependency structure in the data sequence. The intuition is that, thanks to Assumption 2 and the fact that $\bar{Q}_{h+1,t} \in \mathcal{G}_{\mathrm{up}}$, we know that there exists a parameter $\bar{\theta}_{h,t}$ such that $f(\langle \phi(s,a), \bar{\theta}_{h,t}\rangle) = \mathcal{T}_h(\bar{Q}_{h+1,t})(s,a)$. It is easy to verify that $\bar{\theta}_{h,t}$ is the Bayes optimal predictor for the square loss problem in (1), and so with a uniform convergence argument we can expect that $\hat{\theta}_{h,t}$ is close to $\bar{\theta}_{h,t}$, which is our desired conclusion.

There are two subtleties with this argument. First, we want to show that $\bar{\theta}_{h,t}$ and $\hat{\theta}_{h,t}$ are close in a data-dependent sense, to obtain the dependence on the $\Lambda_{h,t}^{-1}$-Mahalanobis norm in the bound. This can be done using vector-valued self-normalized martingale inequalities (Peña et al., 2008), as in prior work on linear stochastic bandits (Abbasi-Yadkori et al., 2012; Filippi et al., 2010; Abbasi-Yadkori et al., 2011).

However, the process we are considering is not a martingale, since $\bar{Q}_{h+1,t}$, which determines the regression targets $y_{h,\tau}$, depends on all data collected so far. Hence $y_{h,\tau}$ is not measurable with respect to the filtration $\mathcal{F}_\tau$, which prevents us from directly applying a self-normalized martingale concentration inequality. To circumvent this issue, we use a uniform convergence argument and introduce a deterministic covering of $\mathcal{G}_{\text{up}}$. Each element of the cover induces a different sequence of regression targets $y_{h,\tau}$, but as the covering is deterministic, we do obtain martingale structure. Then, we show that the error term for the random $\bar{Q}_{h+1,t}$ that we need to bound is close to a corresponding term for one of the covering elements, and we finish the proof with a uniform convergence argument over all covering elements.

The corollary is then obtained by a straightforward inductive argument. Assuming $\bar{Q}_{h+1,t}$ dominates $Q^\star$, it is easy to show that $\bar{Q}_{h,t}$ also dominates $Q^\star$, and the upper bound is immediate. Combining Corollary 4 with Corollary 3, all that remains is to upper bound the confidence sum.

**Potential argument.** To bound the confidence sum, we use a standard potential argument that appears in a number of works on stochastic linear bandits. We summarize the conclusion with the following lemma, which follows directly from Lemma 11 of Abbasi-Yadkori et al. (2012).

**Lemma 3.** *For any $h \leq H$ we have that*

$$\sum_{t=1}^{T} \|\phi(s_{h,t}, a_{h,t})\|_{\Lambda_{h,t-1}^{-1}}^2 \leq 2d\ln(1 + T/d).$$

**Wrapping up.** Equipped with the above results, we are now prepared to prove Theorem 1.

*Proof of Theorem 1.* Assume that Corollary 4 holds for all $1 \leq h \leq H$ and $1 \leq t \leq T$. Applying Lemma 1 and the definition of $\text{cnf}_{h,t-1}$ implied by Corollary 4, the cumulative expected regret is at most

$$TV^\star - \mathbb{E}\left[\sum_{t=1}^{T}\sum_{h=1}^{H} r_{h,t}\right]$$

$$\leq \sum_{t=1}^{T} \zeta_t + \sum_{t=1}^{T}\sum_{h=1}^{H} \min\left\{2, \gamma \|\phi(s_{h,t}, a_{h,t})\|_{\Lambda_{h,t-1}^{-1}}\right\}$$

$$\leq \sum_{t=1}^{T} \zeta_t + \sum_{h=1}^{H} \sqrt{T\gamma^2} \cdot \sqrt{\sum_{t=1}^{T} \|\phi(s_{h,t}, a_{h,t})\|_{\Lambda_{h,t-1}^{-1}}^2}$$

$$\leq \sum_{t=1}^{T} \zeta_t + \sum_{h=1}^{H} \sqrt{T\gamma^2} \cdot \sqrt{2d\ln(1 + T/d)}.$$

Here, the second step follows from the Cauchy-Schwarz inequality, and the last step is an application of Lemma 3. The first term forms a martingale, and we know that $|\zeta_t| \leq 2H$. Therefore, by Azuma's inequality, we have that with probability at least $1 - 1/TH$

$$\sum_{t=1}^{T} \zeta_t \leq \sqrt{8TH^2 \ln(TH)}.$$

Finally, using the definition of $\gamma$, the final regret is upper bounded by

$$\text{Reg}(T) \leq O\big(H\sqrt{T\ln(TH)} + HK\kappa^{-1} \times \sqrt{(M + K + d^2\ln((K + \Gamma)TH)) \cdot Td\ln(1 + T/d)}\big)$$

$$\leq \tilde{O}\left(H\sqrt{d^3 T}\right),$$

which proves the result. $\square$

## B    PROOF OF LEMMA 2 AND COROLLARY 4

To facilitate our analysis we define the following important intermediate quantity:

$$\bar{\theta}_{h,t} \in \mathbb{B}_d : \quad f(\langle \phi(s,a), \bar{\theta}_{h,t} \rangle) \triangleq \mathbb{E}\left[ r_h + \max_{a' \in \mathcal{A}} \bar{Q}_{h+1,t}(s',a') \mid s,a \right].$$

In words, $\bar{\theta}_{h,t}$ is the Bayes optimal predictor for the squared loss problem at time point $h$ in the $t^{\text{th}}$ episode. Since by inspection $\bar{Q}_{h+1,t} \in \mathcal{G}_{\text{up}}$, by Assumption 2 we know that $\bar{\theta}_{h,t}$ exists for all $h$ and $t$.

**Lemma 4.** *For any $\theta, \theta', x \in \mathbb{R}^d$ satisfying $\|\theta\|_2, \|\theta'\|_2, \|x\|_2 \leq 1$,*

$$\kappa^2 |\langle x, \theta' - \theta \rangle|^2 \leq |f(\langle x, \theta' \rangle) - f(\langle x, \theta \rangle)|^2 \leq K^2 \|\theta' - \theta\|_2^2.$$

*Proof.* By the mean-value theorem, there exists $\tilde{\theta} = \theta + \lambda(\theta' - \theta)$ for some $\lambda \in (0,1)$ such that $f(\langle x, \theta' \rangle) - f(\langle x, \theta \rangle) = \left\langle \nabla_\theta f(\langle x, \tilde{\theta} \rangle), \theta' - \theta \right\rangle$. On the other hand, by the chain rule and Assumption 1, $\nabla_\theta f(\langle x, \tilde{\theta} \rangle) = f'(\langle x, \tilde{\theta} \rangle) \cdot x$. Hence,

$$|\langle \nabla_\theta f(x^\top \tilde{\theta}), \theta' - \theta \rangle|^2 \leq f'(\langle x, \tilde{\theta} \rangle)^2 \cdot |\langle x, \theta' - \theta \rangle|^2 \leq K^2 \|x\|_2^2 \|\theta' - \theta\|_2^2 \leq K^2 \|\theta' - \theta\|_2^2;$$
$$|\langle \nabla_\theta f(x^\top \tilde{\theta}), \theta' - \theta \rangle|^2 \geq \kappa^2 |\langle x, \theta' - \theta \rangle|^2,$$

which are to be demonstrated. $\qquad \square$

**Lemma 5.** *For any $0 < \varepsilon \leq 1$, there exists a finite subset $\mathcal{V}_\varepsilon \subset \mathcal{G}_{\text{up}}$ with $\ln |\mathcal{V}_\varepsilon| \leq 6d^2 \ln(2(1 + K + \Gamma)/\varepsilon)$, such that*

$$\sup_{g \in \mathcal{G}_{\text{up}}} \min_{v \in \mathcal{V}_\varepsilon} \sup_{s,a} |g(\phi(s,a)) - v(\phi(s,a))| \leq \varepsilon. \tag{4}$$

*Proof.* Recall that for every $g \in \mathcal{G}_{\text{up}}$, there exists $\theta \in \mathbb{B}_d$, $0 \leq \gamma \leq \Gamma$ and $\|A\|_{\text{op}} \leq 1$ such that $g(x) = \min\{1, f(\langle x, \theta \rangle) + \gamma \|x\|_A\}$. Let $\Theta_\varepsilon \subseteq \mathbb{B}_d$, $\Gamma_\varepsilon \subseteq [0, \Gamma]$ and $\mathcal{M}_\varepsilon \subseteq \{M \in \mathbb{S}_d^+ : \|M\|_{\text{op}} \leq 1\}$ be finite subsets such that for any $\theta, \gamma, A$, there exist $\theta' \in \Theta_\varepsilon$, $\gamma' \in \Gamma_\varepsilon$, $A' \in \mathcal{M}_\varepsilon$ such that

$$\max\left\{ \|\theta - \theta'\|_2, |\gamma - \gamma'|, \|A - A'\|_{\text{op}} \right\} \leq \varepsilon',$$

where $\varepsilon' \in (0,1)$ will be specified later in the proof. For the function $g \in \mathcal{G}_{\text{up}}$ corresponding to the parameters $\theta, \gamma, A$ the function $g'$ corresponding to parameters $\theta', \gamma', A'$ satisfies

$$\begin{aligned}
\sup_{s,a} |g(\phi(s,a)) - g'(\phi(s,a))| &\leq \sup_{x \in \mathbb{B}_d} |g(x) - g'(x)| \\
&\leq \sup_{x \in \mathbb{B}_d} |f(\langle x, \theta \rangle) - f(\langle x, \theta' \rangle) + \gamma \|x\|_A - \gamma' \|x\|_{A'}| \\
&\leq K \|\theta - \theta'\|_2 + |\gamma - \gamma'| + \Gamma \| \|x\|_A - \|x\|_{A'} | \\
&\leq K \|\theta - \theta'\|_2 + |\gamma - \gamma'| + \Gamma \sqrt{|x^\top (A - A')x|} \\
&\leq K\varepsilon' + \varepsilon' + \Gamma \sqrt{\varepsilon'} \leq (1 + K + \Gamma) \sqrt{\epsilon'}.
\end{aligned}$$

In the last step we use $\varepsilon' \leq 1$. Therefore, if we define the class $\mathcal{V}_\varepsilon \triangleq \{(s,a) \mapsto \min\{1, f(\langle \phi(s,a), \theta' \rangle) + \gamma' \|\phi(s,a)\|_{A'} : \theta' \in \Theta_\varepsilon, \gamma \in \Gamma_\varepsilon, A \in \mathcal{M}_\varepsilon\}$, we know that the covering property is satisfied with parameter $(1 + K + \Gamma)\sqrt{\varepsilon'}$. Setting $\varepsilon' = \varepsilon^2/(1 + K + \Gamma)^2$ we have the desired covering property.

Finally, we upper bound $\ln |\mathcal{V}_\varepsilon|$. By definition, we have that $\ln |\mathcal{V}_\varepsilon| \leq \ln |\Theta_\varepsilon| + \ln |\Gamma_\varepsilon| + \ln |\mathcal{M}_\varepsilon|$. Furthermore, standard covering number bounds reveals that $\ln |\Theta_\varepsilon| \leq d \ln(2/\varepsilon')$, $\ln |\Gamma_\varepsilon| \leq \ln(1/\varepsilon')$ and $\ln |\mathcal{M}_\varepsilon| \leq d^2 \ln(2/\varepsilon')$. Plugging in the definition of $\varepsilon'$ yields the result. $\qquad \square$

For the next lemma, let $\mathcal{F}_{t-1} \triangleq \sigma(\{(s_{h,\tau}, a_{h,\tau}, r_{h,\tau})\}_{\tau < t})$ be the filtration induced by all observed trajectories up to but not including time $t$. Observe that $\bar{Q}_{\cdot, t-1}$ and our policy $\hat{\pi}_{h,t}$ are $\mathcal{F}_{t-1}$ measurable.

**Lemma 6** (Restatement of Lemma 2). *Fix any $1 \le t \le T$ and $1 \le h \le H$. Then as long as $\pi_t$ is $\mathcal{F}_{t-1}$ measurable, with probability $1 - 1/(TH)^2$ it holds that*

$$\left| f(\langle \phi(s,a), \hat{\theta}_{h,t} \rangle) - f(\langle \phi(s,a), \bar{\theta}_{h,t} \rangle) \right| \le \min \left\{ 2, \gamma \left\| \phi(s,a) \right\|_{\Lambda_{h,t}^{-1}} \right\}, \qquad \forall s, a.$$

*for $\gamma \ge CK\kappa^{-1}\sqrt{1 + M + K + d^2 \ln((1 + K + \Gamma)TH)}$ and $0 < C < \infty$ is a universal constant.*

Note that this is precisely Lemma 2, as $\bar{\theta}_{h,t}$ is defined as $f(\langle \phi(s,a), \bar{\theta}_{h,t} \rangle) = \mathcal{T}_h(\bar{Q}_{h+1,t})(s,a)$.

*Proof.* The upper bound of 2 is obvious, since both terms are upper bounded by 1 in absolute value. Therefore we focus on the second term in the minimum. To simplify notation we omit the dependence on $h$ in the subscripts and write $x_\tau, y_\tau$ for $x_{h,\tau}$ and $y_{h,\tau}$. We also abbreviate $\hat{\theta} \triangleq \hat{\theta}_{h,t}$ and $\bar{\theta} \triangleq \bar{\theta}_{h,t}$.

Since $\left\| \bar{\theta} \right\|_2 \le 1$, the optimality of $\hat{\theta}$ for (1) implies that

$$\sum_{\tau \le t} \left( f(\langle x_\tau, \hat{\theta} \rangle) - y_\tau \right)^2 \le \sum_{\tau \le t} \left( f(\langle x_\tau, \bar{\theta} \rangle) - y_\tau \right)^2.$$

Decomposing the squares and re-organizing the terms, we have that

$$\sum_{\tau \le t} \left( f(\langle x_\tau, \hat{\theta} \rangle) - f(\langle x_\tau, \bar{\theta} \rangle) \right)^2 \le 2 \left| \sum_{\tau \le t} \xi_\tau (f(\langle x_\tau, \hat{\theta} \rangle) - f(\langle x_\tau, \bar{\theta} \rangle)) \right|, \tag{5}$$

where $\xi_\tau \triangleq y_\tau - f(\langle x_\tau, \bar{\theta} \rangle)$. By the fundamental theorem of calculus, we have

$$f(\langle x_\tau, \hat{\theta} \rangle) - f(\langle x_\tau, \bar{\theta} \rangle) = \int_{\langle x_\tau, \bar{\theta} \rangle}^{\langle x_\tau, \hat{\theta} \rangle} f'(s) \mathrm{d}s = \langle x_\tau, \hat{\theta} - \bar{\theta} \rangle \underbrace{\int_0^1 f'(\langle x_\tau, s\hat{\theta} + (1-s)\bar{\theta} \rangle) \mathrm{d}s}_{\triangleq D_\tau}.$$

Using this identity on both sides of (5), we have that

$$\sum_{\tau \le t} D_\tau^2 \left( \langle x_\tau, \hat{\theta} - \bar{\theta} \rangle \right)^2 \le 2 \left| \sum_{\tau \le t} \xi_\tau D_\tau \langle x_\tau, \hat{\theta} - \bar{\theta} \rangle \right|. \tag{6}$$

Note also that, by Assumption 1, $D_\tau$ satisfies $\kappa^2 \le D_\tau^2 \le K^2$ almost surely for all $\tau$.

The difficulty in controlling (6) is that $\bar{\theta}$ itself is a random variable that depends on $\{(x_\tau, y_\tau)\}_{\tau \le t}$. In particular, we want that $\mathbb{E}[\xi_\tau \mid D_\tau \langle x_\tau, \phi \rangle, \mathcal{F}_{\tau-1}] = 0$ for any fixed $\phi$, but this is not immediate as $\bar{\theta}$ depends on $x_\tau$. To proceed, we eliminate this dependence with a uniform convergence argument. Let $\varepsilon \in (0,1)$ be a covering accuracy parameter to be determined later in this proof. Let $\mathcal{V}_\varepsilon$ be the pointwise covering for $\mathcal{G}_{\mathrm{up}}$ that is implied by Lemma 5. Let $g_\varepsilon \in \mathcal{V}_\varepsilon$ be the approximation for $\bar{Q}_{h+1,t}$ that satisfies (4). By Assumption 2, there exists some $\theta^\sharp \in \mathbb{B}_d$ such that

$$\forall s, a: \quad f(\langle \phi(s,a), \theta^\sharp \rangle) = \mathbb{E}\left[ r + \max_{a' \in \mathcal{A}} g_\varepsilon(s', a') \mid s, a \right].$$

Now, define $y_\tau^\sharp$ and $\xi_\tau^\sharp$ as

$$y_\tau^\sharp \triangleq r_{h,\tau} + \max_{a' \in \mathcal{A}} g_\varepsilon(s_{h+1,\tau}, a'), \qquad \xi_\tau^\sharp \triangleq y_\tau^\sharp - f(\langle x_{h,\tau}, \theta^\sharp \rangle).$$

The right-hand side of (6) can then be upper bounded as

$$2 \left| \sum_{\tau \le t} \xi_\tau D_\tau \langle x_\tau, \hat{\theta} - \bar{\theta} \rangle \right| \le 2 \left| \sum_{\tau \le t} \xi_\tau^\sharp D_\tau \langle x_\tau, \hat{\theta} - \bar{\theta} \rangle \right| + \Delta, \tag{7}$$

where $|\Delta| \le Kt \times \max_{\tau \le t} |\xi_\tau^\sharp - \xi_\tau|$ almost surely.

**Upper bounding $\Delta$ in (7).** Fix $\tau \le t$. By definition, we have that

$$
\begin{aligned}
\left| \xi_\tau^\sharp - \xi_\tau \right| &\le \left| y_\tau^\sharp - y_\tau \right| + \left| f(\langle x_\tau, \bar{\theta} \rangle) - f(\langle x_\tau, \theta^\sharp \rangle) \right| \\
&\le \max_{a \in \mathcal{A}} \left| g_\varepsilon(s_{h+1,\tau}, a) - \bar{Q}_{h+1,t}(s_{h+1,\tau}, a) \right| + K \left\| \bar{\theta} - \theta^\sharp \right\|_2 \qquad (8) \\
&\le \epsilon + K\epsilon \le (K+1)\epsilon, \qquad (9)
\end{aligned}
$$

where (8) holds by Lemma 4 and (9) follows from Lemma 5. In particular, the bound on $\left\| \bar{\theta} - \theta^\sharp \right\|_2$ can be verified by expanding the definitions and noting that $g_\varepsilon$ is pointwise close to $\bar{Q}_{h+1,t}$. Therefore, we have

$$
|\Delta| \le (K+1)^2 t\epsilon. \qquad (10)
$$

**Upper bounding (7).** Note that $D_\tau$ is a function of $x_\tau$, $\hat{\theta}$, and $\bar{\theta}$. For clarity, we define $D_\tau(\theta, \theta') := \int_0^1 f'(\langle x_\tau, s\theta + (1-s)\theta' \rangle)\mathrm{d}s$. As $|f''(z)| \le M$ for all $|z| \le 1$ and $\|x_\tau\|_2 \le 1$, we have that for every $\theta, \theta', \tilde{\theta}, \tilde{\theta}' \in \mathbb{B}_d$

$$
\begin{aligned}
\left| D_\tau(\theta, \theta') - D_\tau(\tilde{\theta}, \tilde{\theta}') \right| &\le \int_0^1 \left| f'(\langle x_\tau, s\theta + (1-s)\theta' \rangle) - f'(\langle x_\tau, s\tilde{\theta} + (1-s)\tilde{\theta}' \rangle) \right| \mathrm{d}s \\
&\le M(\|\theta - \tilde{\theta}\|_2 + \|\theta' - \tilde{\theta}'\|_2).
\end{aligned}
$$

Hence, for any $(\theta, \theta')$ and $(\tilde{\theta}, \tilde{\theta}')$ pairs, we have for every $\tau$ that

$$
\begin{aligned}
&\left| \xi_\tau^\sharp \left\langle x_\tau, D_\tau(\theta, \theta')(\theta - \theta') - D_\tau(\tilde{\theta}, \tilde{\theta}')(\tilde{\theta} - \tilde{\theta}') \right\rangle \right| \\
&\le \left| D_\tau(\theta, \theta') - D_\tau(\tilde{\theta}, \tilde{\theta}') \right| \times \|\theta - \theta'\|_2 + \left| D_\tau(\tilde{\theta}, \tilde{\theta}') \right| \times (\|\theta - \tilde{\theta}\|_2 + \|\theta' - \tilde{\theta}'\|_2) \\
&\le M(\|\theta - \tilde{\theta}\|_2 + \|\theta' - \tilde{\theta}'\|_2) \times 2 + K(\|\theta - \tilde{\theta}\|_2 + \|\theta' - \tilde{\theta}'\|_2) \\
&\le (2M + K)(\|\theta - \tilde{\theta}\|_2 + \|\theta' - \tilde{\theta}'\|_2).
\end{aligned}
$$

Here we are using that $|\xi_\tau| \le 1$.

We are now in a position to invoke Lemma 8. Consider a fixed function $g_\varepsilon$, which defines a fixed $\theta^\sharp$. We will bound $\left| \sum_{\tau \le t} \xi_\tau^\sharp \langle x_\tau, D_\tau(\theta, \theta')(\theta - \theta') \rangle \right|$ uniformly over all pairs $(\theta, \theta')$. With $g_\varepsilon, \theta^\sharp$ fixed and since $\pi_t$ is $\mathcal{F}_{t-1}$ measurable, we have that $\{x_\tau, \xi_\tau^\sharp\}_{\tau \le t}$ are random variables satisfying $\mathbb{E}[\xi_\tau^\sharp \mid x_{1:\tau}, \xi_{1:\tau-1}^\sharp] = 0$. For $\phi = (\theta, \theta')$ we define the function $q(x_\tau, \phi) = \langle x, D_\tau(\phi)(\theta - \theta') \rangle$, which as we have just calculated satisfies $|q(x_\tau, \phi) - q(x_\tau, \phi')| \le (2M + K)\|\phi - \phi'\|_2$. For $\delta' \in (0, 1/2)$ with probability $1 - \delta'$ we have $\forall \phi = (\theta, \theta') \in \mathbb{B}_d^2$:

$$
\begin{aligned}
\left| \sum_{\tau \le t} \xi_\tau^\sharp \langle x_\tau, D_\tau(\phi)(\theta - \theta') \rangle \right| &\le (2M + K) + 2\left(1 + \sqrt{V(\phi)}\right)\sqrt{2d\ln(4T) + \ln(1/\delta')} \\
&\le 4\max\left\{ M + K + \sqrt{2d\ln(4T) + \ln(1/\delta')}, \sqrt{V(\phi)}\sqrt{2d\ln(4T) + \ln(1/\delta')} \right\}, \qquad (11)
\end{aligned}
$$

where $V(\phi) \triangleq \sum_{\tau \le t} \langle x_\tau, D_\tau(\phi)(\theta - \theta') \rangle^2$. The last inequality holds because $a + b \le 2\max\{a, b\}$.

Next, take a union bound over all $g_\varepsilon \in \mathcal{V}_\varepsilon$ so (11) holds for any $g_\varepsilon$ and any subsequently induced choice of $\xi_\tau^\sharp$ with probability at least $1 - |\mathcal{V}_\varepsilon|\delta'$. In particular, this union bound implies that (11) holds for the choice of $g_\varepsilon$ that approximates $\bar{Q}_{h+1,t}$. Therefore, combining (6), (7), (10) with (11) for this choice of $g_\varepsilon$, we have that with probability at least $1 - |\mathcal{V}_\varepsilon|\delta'$

$$
\begin{aligned}
\sum_{\tau \le t} D_\tau^2 \langle x_\tau, \hat{\theta} - \bar{\theta} \rangle^2 &\le 2\Delta + 2\left| \sum_{\tau \le t} \xi_\tau^\sharp \langle x_\tau, D_\tau(\hat{\theta} - \bar{\theta}) \rangle \right| \\
&\le 2(K+1)^2 t\varepsilon + 8\max\left\{ M + K + \sqrt{2d\ln(4T) + \ln(|\mathcal{V}_\varepsilon|/\delta')}, \sqrt{V(\hat{\theta}, \bar{\theta})} \cdot \sqrt{2d\ln(4T) + \ln(|\mathcal{V}_\varepsilon|/\delta')} \right\}.
\end{aligned}
$$

Observe that the left hand side is precisely $V(\hat{\theta}, \bar{\theta})$. Now, set $\varepsilon = 1/(2(K+1)^2 T)$ and $\delta' = 1/(|\mathcal{V}_\varepsilon|T^2 H^2)$ and use the bound on $\ln|\mathcal{V}_\varepsilon|$ from Lemma 5 to get

$$\sqrt{2d\ln(4T) + \ln(|\mathcal{V}_\varepsilon|/\delta')} \leq \sqrt{2d\ln(4T) + 12d^2\ln(2(1+K+\Gamma)/\varepsilon) + 2\ln(TH)}$$
$$\leq \sqrt{4d\ln(2TH) + 24d^2\ln(2(1+K+\Gamma)T)} \leq \sqrt{28d^2\ln(2(1+K+\Gamma)TH)}$$

Therefore, we obtain

$$V(\hat{\theta}, \bar{\theta}) \leq 1 + 8\max\left\{M + K + \sqrt{28d^2\ln(2(1+K+\Gamma)TH)}, \sqrt{V(\hat{\theta}, \bar{\theta})} \cdot \sqrt{28d^2\ln(2(1+K+\Gamma)TH)}\right\}$$
$$\leq 16\max\left\{1 + M + K + \sqrt{28d^2\ln(2(1+K+\Gamma)TH)}, \sqrt{V(\hat{\theta}, \bar{\theta})} \cdot \sqrt{28d^2\ln(2(1+K+\Gamma)TH)}\right\}.$$

Subsequently,

$$V(\hat{\theta}, \bar{\theta}) = \sum_{\tau \leq t} D_\tau^2 \langle x_\tau, \hat{\theta} - \bar{\theta}\rangle^2$$
$$\leq 16\max\left\{1 + M + K + \sqrt{28d^2\ln(2(1+K+\Gamma)TH)}, 448d^2\ln(2(1+K+\Gamma)TH)\right\}$$
$$\leq C_V^2(1 + M + K + d^2\ln((1+K+\Gamma)TH)),$$

where $0 < C_V < \infty$ is a universal constant.

Next, note that $D_\tau^2 \geq \kappa^2$, thanks to Assumption 1. We then have

$$\sqrt{(\hat{\theta} - \bar{\theta})^\top \Lambda_{h,t}(\hat{\theta} - \bar{\theta})} \leq \kappa^{-1}\sqrt{V(\hat{\theta}, \bar{\theta})} \leq C_V \kappa^{-1}\sqrt{1 + M + K + d^2\ln((1+K+\Gamma)TH)},$$

where $\Lambda_{h,t} = \sum_{\tau < t} x_\tau, x_\tau^\top$. Finally, for any $(s, a)$ pair, invoking Lemma 4 and the Cauchy-Schwarz inequality we have

$$\left|f(\langle\phi(s,a), \hat{\theta}\rangle) - f(\langle\phi(s,a), \bar{\theta}\rangle)\right| \leq K\left|\langle\phi(s,a), \hat{\theta} - \bar{\theta}\rangle\right|$$
$$\leq K\sqrt{(\hat{\theta} - \bar{\theta})^\top \Lambda_{h,t}(\hat{\theta} - \bar{\theta})} \times \sqrt{\phi(s,a)^\top \Lambda_{h,t}^{-1}\phi(s,a)}$$
$$\leq C_V K\kappa^{-1}\sqrt{1 + M + K + d^2\ln((1+K+\Gamma)TH)} \times \|\phi(s,a)\|_{\Lambda_{h,t}^{-1}}$$

which is to be demonstrated. $\qquad\square$

**Corollary 5** (Restatement of Corollary 4). *With probability $1 - 1/(TH)$, $\bar{Q}_{h,t}(s,a) \geq Q_h^\star(s,a)$ holds for all $h, t, s, a$.*

*Proof.* Fix $1 \leq t \leq T$. We use induction on $h$ to prove this corollary. For $h = H+1$, $\bar{Q}_{H+1,t}(\cdot, \cdot) \geq Q_{H+1}^\star(\cdot, \cdot)$ clearly holds because $\bar{Q}_{H+1,t} \equiv Q_{H+1}^\star \equiv 0$. Now assume that $\bar{Q}_{h+1,t} \geq Q_{h+1}^\star$, and let us prove that this is also true for time step $h$.

Since $\bar{Q}_{h+1,t}(s', a') \geq Q_{h+1}^\star(s', a')$ for all $s', a'$, we have that $f(\langle\phi(s,a), \bar{\theta}_{h,t}\rangle) \geq f(\langle\phi(s,a), \theta_h^\star\rangle)$ for all $(s, a)$ pairs. Then, by the definition of $\bar{Q}_{h,t}$ and Lemma 6, with probability $1 - 1/(TH)^2$ it holds uniformly for all $(s, a)$ pairs that $\bar{Q}_{h,t}(s,a) \geq f(\langle\phi(s,a), \bar{\theta}_{h,t}\rangle)$. Hence, with the same probability, we have $\bar{Q}_{h,t}(s,a) \geq Q_h^\star(s,a)$ for all $(s, a)$ pairs. A union bound over all $t \leq T$ and $h \leq H$ completes the proof. $\qquad\square$

## C  TAIL INEQUALITIES

**Lemma 7** (Azuma's inequality). *Suppose $X_0, X_1, X_2, \cdots, X_N$ form a martingale (i.e., $\mathbb{E}[X_{k+1}|X_1, \cdots, X_k] = X_k$) and satisfy $|X_k - X_{k-1}| \leq c_k$ almost surely. Then for any $\epsilon > 0$,*

$$\Pr\left[\left|X_n - X_0\right| \geq \epsilon\right] \leq 2\exp\left\{-\frac{\epsilon^2}{2\sum_{k=1}^N c_k^2}\right\}.$$

**Lemma 8.** *Fix* $t, D \in \mathbb{N}$. *Let* $\{\xi_\tau, u_\tau\}_{\tau \leq t}$ *be random variables such that* $\mathbb{E}[\xi_\tau | u_1, \xi_1, \cdots, u_{\tau-1}, \xi_{\tau-1}, u_\tau] = 0$ *and* $|\xi_\tau| \leq 1$ *almost surely. Let* $q : (u, \phi) \mapsto \mathbb{R}$ *be an arbitrary deterministic function satisfying* $|q(u, \phi) - q(u, \phi')| \leq C\|\phi - \phi'\|_2$ *for all* $u, \phi$ *and* $\phi'$, *where* $\phi, \phi' \in \mathbb{R}^D$. *Then for any* $\delta \in (0, 1)$ *and* $R > 0$,

$$\Pr\left[\forall \phi \in \mathbb{B}_D(R) : \left|\sum_{\tau=1}^{t} \xi_\tau q(u_\tau, \phi)\right| \leq C + 2\left(1 + \sqrt{V_q(\phi)}\right)\sqrt{D\ln(2tR) + \ln(1/\delta)}\right] \geq 1 - \delta,$$

*where* $\mathbb{B}_D(R) \triangleq \{x \in \mathbb{R}^D : \|x\|_2 \leq R\}$ *and* $V_q(\phi) \triangleq \sum_{\tau \leq t} q^2(u_\tau, \phi)$.

*Proof.* Let $\epsilon > 0$ be a small precision parameter to be specified later. Let $\mathcal{H} \subseteq \mathbb{B}_D(R)$ be a finite $\epsilon$-covering of $\mathbb{B}_D(R)$ such that $\sup_{x \in \mathbb{B}_D(R)} \min_{z \in \mathcal{H}} \|x - z\|_2 \leq \epsilon$. Using standard covering number arguments, such a covering exists with $\ln |\mathcal{H}| \leq D\ln(2R/\epsilon)$.

For any $\phi \in \mathbb{B}_D(R)$ let $\phi' \triangleq \operatorname{argmin}_{z \in \mathcal{H}} \|\phi - z\|_2$. By definition, $\|\phi - \phi'\|_2 \leq \epsilon$. This implies $\left|\sum_{\tau=1}^{t} \xi_\tau [q(u_\tau, \phi) - q(u_\tau, \phi')]\right| \leq Ct\epsilon$ because $|\xi_\tau| \leq 1$ almost surely. Subsequently, for any $\Delta > 0$,

$$\Pr\left[\exists \phi \in \mathbb{B}_D(R) : \left|\sum_{\tau=1}^{t} \xi_\tau q(u_\tau, \phi)\right| > Ct\epsilon + \Delta\right] \leq \Pr\left[\exists \phi' \in \mathcal{H} : \left|\sum_{\tau=1}^{t} \xi_\tau q(u_\tau, \phi')\right| > \Delta\right]$$

$$\leq \sum_{\phi' \in \mathcal{H}} \Pr\left[\left|\sum_{\tau=1}^{t} \xi_\tau q(u_\tau, \phi')\right| > \Delta\right],$$

where the last inequality holds by the union bound.

For any fixed $\phi' \in \mathcal{H}$, $h(u_\tau, \phi')$ only depends on $u_\tau$, and therefore $\mathbb{E}[\xi_\tau \mid q(u_\tau, \phi')] = 0$ for all $\tau$. Invoking Lemma 7 with $X_\tau \triangleq \sum_{\tau' \leq \tau} \xi_{\tau'} q(u_{\tau'}, \phi')$ and $c_{\tau'} = |q(u_{\tau'}, \phi')|$, we have

$$\Pr\left[\left|\sum_{\tau=1}^{t} \xi_\tau q(u_\tau, \phi')\right| > \Delta\right] \leq 2\exp\left\{\frac{-\Delta^2}{2\sum_{\tau \leq t} q^2(u_\tau, \phi')}\right\} = 2\exp\left\{\frac{-\Delta^2}{2V_q(\phi')}\right\}$$

Equating the right-hand side of the above inequality with $\delta'$ and combining with the union bound application, we have

$$\Pr\left[\exists \phi \in \mathbb{B}_d(R) : \left|\sum_{\tau=1}^{t} \xi_\tau h(u_\tau, \phi)\right| > Ct\epsilon + \sqrt{2V_q(\phi')\ln(2/\delta')}\right] \leq \delta'|\mathcal{H}|. \qquad (12)$$

Further equating $\delta' = \delta/|\mathcal{H}|$ and using the fact that $\ln |\mathcal{H}| \leq D\ln(2R/\epsilon)$, we have

$$\Pr\left[\exists \phi \in \mathbb{B}_d(R) : \left|\sum_{\tau=1}^{t} \xi_\tau q(u_\tau, \phi)\right| > Ct\epsilon + \sqrt{2DV_q(\phi')\ln(2R/\epsilon) + 2V_q(\phi')\ln(1/\delta)}\right] \leq \delta.$$

Finally, as $|q(u_\tau, \phi') - q(u_\tau, \phi)| \leq \epsilon$, we have $V_q(\phi') \leq 2V_q(\phi) + 2t\epsilon^2$ and so

$$\Pr\left[\exists \phi \in \mathbb{B}_D(R) : \left|\sum_{\tau=1}^{t} \xi_\tau q(u_\tau, \phi)\right| > Ct\epsilon + 2\epsilon\sqrt{Dt\ln(2R/\epsilon\delta)} + 2\sqrt{V_q(\phi)(D\ln(2R/\epsilon) + \ln(1/\delta))}\right] \leq \delta.$$

Setting $\epsilon = 1/t$ in the above inequality completes the proof. $\qquad \square$

