# OpenReview forum: "Optimism in Reinforcement Learning with Generalized Linear Function Approximation"
_ICLR.cc/2021/Conference — ICLR 2021 Poster_

### Official Review · AnonReviewer2 · 2020-10-27
**A nice extenstion of analysis for LSVI-UCB with generalized linear function approximation**

**Rating:** 6
**Confidence:** 4

**Review:**

### Summary
This paper analyses an existing algorithm (LSVI-UCB) with generalized linear function approximation instead of conventional linear function approximation.  Under this generalized linear setting, they propose a so-called “optimistic closure” assumption which is shown to be strictly weaker than the expressivity assumption in the conventional linear setting. The paper then proves that LSVI-UCB still enjoys sub-linear regret in the generalized linear setting with strictly weaker assumptions. The paper also derives a general error propagation through steps that do not require a closed-form expression of the empirical dynamic and reward functions as in the linear case; this could be applicable to general function approximations.
### Strong points
-	Novelty: The generalized linear setting appears novel and generalizes the linear settings.
-	Significance: The optimistic closure appears novel and is strictly weaker than the linear MDP assumption in the prior works.
-	Correctness: A complete analysis that successfully retains a sublinear regret and honest comments on the limitations of the present work.

### Weak points
-	The work is almost merely about analysis of an existing algorithm with modest algorithmic contribution (which however is not a big problem). There are some parts of the proofs pointed out in the Minor comment section that potentially require some attention (but I believe these are minor points which could be fixed if there is any issue)

### Minor comments
-	Period ‘.’ after the first sentence of the second paragraph of section 2.
-	First sentence of section 3: ‘MPD’ -> ‘MDP’
-	Lemma 1: Should it be $\pi_{h,t}$ instead of $\pi_t$ there?
-	In Appendix A: “We believe these technical results will be useful in designing RL algorithms for general function classes”. It seems that an analysis of LSVI-UCB with general function classes has recent done in [1] (?)
-	In Corollary 4, shouldn’t it be **2** $\gamma \| \phi(s,a) \|$ instead of $\gamma \| \phi(s,a) \|_{…}$?
-	At the end of Page 12: “The first term forms a martingale” -> shouldn’t it be a “difference martingale” instead?
-	The equation between eq. (5) and eq. (6) on page 14 does not look very right. I think the correct one should be the one with the RHS replaced by $\langle x_{\tau}, \hat{\theta} - \bar{\theta} \rangle f’(\langle x_{\tau}, s \hat{\theta} + (1-s) \bar{\theta} \rangle)$ for some $s \in [0,1]$ (according to the mean value theorem). If this is true, I am afraid the bounds of the difference between $D_{\tau}$ (after Eq. (10)) might not be precise.
-	The second paragraph on page 12: “Hence $y_{h, \tau}$ is not measurable with
respect to the filtration $F_{\tau}$ ,  which prevents us from directly applying a self-normalized martingale
concentration inequality”. Should it be $F_{\tau-1}$ instead of $F_{\tau}$?

-	On page 15, the paper says that E[   xi_tau^# | x_{1:tau}, xi_{1 : tau-1}^#   ] = 0. Do we really need that martingale structure when we already consider a fixed g_{epsilon}? Given a fixed g_{\epsilon}, we already have E[   xi_tau^# | x_tau   ] = 0.


### Questions for the authors
-	In Chi Jin et al. 2019, the regret is the difference between the optimal value function and the value function estimate while in the present paper, the regret is the difference between the optimal value function and the expected value of the cumulative rewards by the algorithm. What is the difference between these two notions of regret? Can it make the two results comparable?
-	In the proof of ‘Fact 1’, why Q^*_H \in \mathcal{G}? For that to hold, it seems to require that the expected reward \mathbb{E}[r_H] has a generalized linear form of \mathcal{G}? If so, one way to fix it is maybe letting 1 <= h <= H (instead of 1 <= h < H) in Assumption 2?
-	It seems that [1] already analyses LSVI-UCB with general function approximations which means that [1] is more general than the present work (?) If so, could the authors comment on the benefit of this work for a generalized linear function class given that an analysis for a general function class has been done? For example, does the present work give a tighter bound when considering generalized linear function as compared to the bound for a general function class in [1]?


### My initial recommendation
Overall, I vote for accepting. An extension from linear settings to generalized linear settings is novel and natural, and it must be done at some point. I think this work is nice for filling in that gap.

### My final recommendation

I remain my initial score after the discussion.

### References
[1] Ruosong Wang et al. “Reinforcement Learning with General Value Function Approximation: Provably Efficient Approach via Bounded Eluder Dimension”


### Additional comments about the correctness of the proof of Lemma 8

I have recently checked their proof of Lemma 8 and noticed one thing that looks a bit strange to me. Since the discussion is over, I hope the authors will clarify/fix it in their final paper. That is, in the proof of Lemma 8 in the step where they applied Lemma 7 (Azuma's),  they used $c_{\tau'} = |q(u_{\tau'}, \phi')|$, but the Azuma's inequality requires that $c_{\tau'}$ is a constant while here $|q(u_{\tau'}, \phi')|$ is a random variable (depending on the random variable $u_{\tau'}$). How is this possible to apply Azuma's inequality here when $c_{\tau'} = |q(u_{\tau'}, \phi')|$ is random?

---

> ### Author Response · Authors · 2020-11-16
> **author response**
>
> We thank the reviewer very much for his/her helpful suggestions. Below we respond to the main concerns/questions from the reviewer.
>
> 1. Difference in regret definition with Jin et al: The objective in Jin et al., is *not* the value function estimate, but rather the true value function of the deployed policy \pi_k. Due to the fact that we are taking an expectation (and we do not update the policy during the episode), the two objectives are actually the same. Note also that the actual collected rewards differ from this quantity by at most H\sqrt{T} due to Azuma-Hoeffding.
>
> 2. In the proof of Fact 1, why Q_H^\star is in \mathcal{G}? Indeed, it is required (as the base case of the inductive proof) that the Q_H^* for the last episode belong to the function class G. We will clarify this in the revised paper, as suggested by the reviewer by strengthening the Assumption 2 to make sure that Q_H^* is in \mathcal{G}.
>
> 3. Comparison to the results of Ruosong Wang et al. We would like to clarify that, the paper mentioned by the reviewer is actually a *follow-up* paper of our results. Indeed, our paper was arxived in December, 2019 and the paper of Ruosong Wang et al. was arxived in May, 2020, in which they clearly cited our paper as a starting point/prior literature.

---

> > ### Comment · AnonReviewer2 · 2020-11-17
> > **Thanks the author response. Please address some of my questions in the "Minor Comment" section as well**
> >
> > I thank the authors for your response. I would like the authors to respond to some of the questions in the minor comments as well before I make my final recommendation. I put them as minor comments because I assume that these technical minors should not affect the main conclusions and theorem of the paper, but they might improve the paper and I would like to see how the authors will take these. For convenience, I will explicitly rewrite some least minor questions here that I would like to hear a response to.
> > -	The equation between eq. (5) and eq. (6) on page 14 does not look very right. I think the correct one should be the one with the RHS replaced by $\langle x_{\tau}, \hat{\theta} - \bar{\theta} \rangle f’(\langle x_{\tau}, s \hat{\theta} + (1-s) \bar{\theta} \rangle)$ for some $s \in [0,1]$ (according to the mean value theorem). If this is true, I am afraid the bounds of the difference between $D_{\tau}$ (after Eq. (10)) might not be precise.
> > -	In Corollary 4, shouldn’t it be **2** $\gamma \| \phi(s,a) \|$ instead of $\gamma \| \phi(s,a) \|_{…}$?
> > -	On page 15, the paper says that E[   xi_tau^# | x_{1:tau}, xi_{1 : tau-1}^#   ] = 0. Do we really need that martingale structure when we already consider a fixed g_{epsilon}? Given a fixed g_{\epsilon}, we already have E[   xi_tau^# | x_tau   ] = 0.

---

> > > ### Author Response · Authors · 2020-11-23
> > > **Thanks for bringing up the minors**
> > >
> > > Thanks for bringing up the minors again, here are some responses:
> > >
> > > 1. There are two typos here and thank you for noticing them. In the middle term, we are integrating the derivative f'(s) instead of f(s); in the last term the integrating term should be $f'(<x_\tau, s\hat\theta+(1-s)\bar\theta>)$. This chain of equalities actually holds by the fundamental theorem of calculus, instead of the mean value theorem.
> > >
> > > 2. Yes you are right, there should be a factor of 2 here.
> > >
> > > 3. Both are correct. We can mention this in the final version.
> > >
> > > We'll make the appropriate changes in the final version. Thanks!

---

### Official Review · AnonReviewer1 · 2020-10-31
**An interesting contributions to the line of research on episodic MDP learning with function approximation**

**Rating:** 7
**Confidence:** 3

**Review:**

The authors studies an episodic MDP learning problem, where they propose to study an Optimistic Closure assumption which allows the Q function to be expressed as a generalized linear function plus a positive semi-definite quadratic form. They motivate the assumption by showing that the assumption allows the tabular MDP case to be modeled, and that the Optimistic Closure is in fact a strictly weaker assumption than the linear MDP assumption made in previous related works. The authors then proceed to the design and analysis of the LSVI-UCB algorithm, which involves estimating the the parameter of the GLM model by a ridge estimator and adding an optimistic exploration bonus to the Q function. The authors propose a regret bound for the algorithm.

The proposed work is an interesting development to the line of research on RL with function approximation, and is large well written. I am in favor of acceptance, given that it provides a non-trivial extension to what is known and the Optimistic Closure assumption seems to me to be closer to the reality than the linear MDP assumption. One suggestion is to investigate if other large scaled but structured MDP models, such as the Factored MDP model by Osband and Van Roy 2014 : https://papers.nips.cc/paper/5445-near-optimal-reinforcement-learning-in-factored-mdps, and the LQG model, satisfy the Optimistic closure assumption with appropriate choices of $\phi, f$.


Minor comments:

In the abstract, brackets are missing for the d^3\sqrt{T} regret bound.

On page 3, $\Gamma$ should be replaced by $\gamma$.

On page 5, MPD -> MDP

---

> ### Author Response · Authors · 2020-11-16
> **Author response**
>
> We thank the reviewer very much for his/her appreciation of our paper and the helpful comments. We do not think that the Factored MDP model satisfies optimistic closure for any "small" Q-function class. Indeed, it is known that the optimal Q function for a factored MDP is in general extremely complicated (formally cannot be expressed as a polynomially sized circuit), and for this reason, all known provably efficient approaches for Factored MDPs are model based. We do not expect model-free methods to be sample-efficient in these environments.
>
> Regarding the LQG, we do know that the simpler LQR satisfies completeness using quadratic value functions, but unfortunately we do not believe it satisfies optimistic completeness. The reason is that with a quadratic value function at time h+1, the one-step optimal policy is linear, which results in a quadratic value function at time h. But the optimism bonus results in a quartic value function at time h+1, which does not admit a closed form optimal policy.

---

### Official Review · AnonReviewer3 · 2020-11-05
**Important generalization; questions on novelty of techniques.**

**Rating:** 6
**Confidence:** 3

**Review:**

The backdrop for this work is the linear MDP model. In linear MDPs, typically the transition function is assumed to be a low rank matrix in the span of d feature vectors (over S, A); such an assumption lends itself to regret bounds that only scale with d (and not explicitly with the size of the state space).

The first contribution here is to establish that it is enough to assume that the function approximation class (for Q functions) is closed under an optimistic (~inverse covariance bonus) version of Bellman update. Qualitatively, this is desirable because this is an assumption on the Q-function class and does not present an explicit assumption on dynamics, unlike linear MDPs. The paper establishes that this is strictly more general the linear MDP assumption, where the above-discussed closure holds for backups of all functions (and not just linear Q functions). It must be noted that Jin et al had already noted & observed that such an assumption is enough, and that their proofs accommodate this.

The second contribution is that the Q function class is generalized here to accommodate generalized (vs just) linear models.

Strengths:
+ I think this is an important relaxation in assumptions to point out. Bellman closure of the policy class seems like a necessary precondition; optimistic variant is a bit further, yet more palatable than a factorization of the dynamics matrix.
+ The GLM part of the extension could be significant in practice, given similar observations in supervised learning.
+ The proof exposition (Appendix A) here is potentially cleaner that Jin et al.

Comments:
+ Regarding the first contribution, did the authors think it was necessary to modify any part of the proof from Jin et al? From my reading, since all concentration arguments were always made on backups, it seemed their proof did indeed go through.
+ Regarding the second contribution, what changes did does this work introduce to handle GLMs? I understand part of the answer may be in Lemma 6.
+ Typo: Page 5 > linear MPD?

---

> ### Author Response · Authors · 2020-11-16
> **Author response**
>
> We thank the reviewer very much for his/her helpful suggestions.
>
> It seems the main concern of this reviewer is regarding the contribution of this paper relative to the results from Jin et al. As remarked by the reviewer, Jin et al's proof goes through as is for linear functions, as they only back up functions in our class \Gcal_{up}. However, as remarked, we feel this is a conceptual point worth emphasizing as the linear MDP does not naturally accommodates the GLM structure.
>
> On the technical side, Our analysis has some differences with that of Jin et al., to address the GLM setting. For example, we use the constrained least squares objective, rather than the regularized objective. This manifests in lemma 6, where we also incorporate the required changes to address GLMs.

---

### Official Review · AnonReviewer5 · 2020-11-06
**A promising line of research, but assumptions are not well motivated and paper isn't clearly written**

**Rating:** 5
**Confidence:** 3

**Review:**

Summary After Discussion Period:
-----------------------------------------------
After corresponding to the authors and reading other reviews, my assessment hasn't changed much, which is that the paper is a good line of research but still needs improvement readability and strictness of assumptions.

The authors and reviewers all point out that this work is a relaxation over some previous works, e.g. Jin et al. Yet [1] has assumptions which are relaxed further than in this paper, and show that at regret bounds are possible with weaker assumptions.

The author's correctly point out their algorithm is computational efficiency while [1]'s algorithm isn't, which is a point in favor of the author's algorithm. Unfortunately, the benefits in computational efficiency were not clear to me and none of the other reviewers highlighted computational efficiency as one of the algorithm's strengths. If indeed one of the author's algorithm's main advantage over other work wasn't clear to the reviewers, then the paper still has room for improvement in readability.

Summary:
--------
In this paper, the authors propose a Q-learning method to solve episodic RL problems. Key to their method is assuming the Q-function takes the form of a generalized linear function plus an optimism term. Once this assumption has been made, they demonstrate that their algorithm, The LSVI-UCB algorithm provably finds a policy with bounded regret.

Pros:
-----
I like that the authors were able to show that using generalized linear functions for the Q function opens the doors to many theoretical analysis possibilities, and I like the modification they made to allow the Q-function to be optimistic. I thought a further strong point of the paper was the proof which shows that using general linearized q functions isn't a stricter assumption than the linear MDP assumption.

And in general, I like the idea. I think it's a good line of research, and an idea which will yield important progress in the field of RL research.

Cons:
-----
I found this paper hard to follow. After multiple readings, I was still confused in multiple areas. This included

  - What is the motivation of the function set $G_\text{op}$?
  - What are some examples of common RL problems for which the Q-function is / is not a generalized linear function
  - Where does the matrix $A$ come from in $G_\text{op}$?
  - What is the motivation for the $\Lambda_{h,t}$ in the algorithm
  - Links to previous work, for example using the generalized linear models as a Q function, it's unclear if this is a new idea or is already present in previous work.
  - It would be good to point out links not just in the related work section, but also while introducing concepts.

To discover why the author's algorithm is optimistic, we need to look at the details of the LSVI-UCB algorithm, a clear explanation isn't given anywhere else.


The other issue I had was with assumption 2. It is a fairly strong assumption, and although the author's show it holds for the linear MDP setting, it isn't nicely motivated why this assumption is realistic for other settings.

In any modeling setting, there's always a bias-variance tradeoff. As the model becomes more complex, it better captures the observations but is more prone to fitting the noise. By assuming the model can perfectly fits the true Q-function, the author's have assumed there's no "bias" in this bias variance tradeoff, and it is not surprising that they can then report lower regret as compared to other methods.

I feel a better analysis should be more along the lines of [1], where they introduce the "inherent bellman error," an error stating how far the true Q function is from the best estimate. One sees that this inherent bellman error then factors prominently in the regret bounds they show. There, they recognize that such a bellman error is generally non-zero, and prove their results by splitting the regret into an approximation and a variance term (like the bias variance trade-off).


Minor concern:
---------------------
In theorem 1, you state the regret is $O(H\sqrt{d^3T})$ while in the abstract it's $O(\sqrt{d^3T})$.
Please keeps it consistent.

In the bibliography, many of the works have been published. It's nice to cite the published version (i.e. the ICML or NeurIPS version) instead of the Arxiv version.

Conclusion:
----------------
If the authors could better address assumption 2 (ideally by doing an analysis akin to [1]), this
would make the theory a good contribution to RL research. And if the authors could write their paper to tell a compelling story, where the different facts, assumptions, definitions and theorems nicely flow into one another, and one understands where things are coming from and where they are going, then this would be a good submission. But in it's current form, with an assumption which masks a large source of regret and a story which is hard to follow, I don't believe this paper is ready for submission.

[1] Andrea Zanette, Alessandro Lazaric, Mykel Kochenderfer, and Emma Brunskill. Learning near
optimal policies with low inherent bellman error. arXiv preprint arXiv:2003.00153, 2020a

---

> ### Author Response · Authors · 2020-11-16
> **Author response**
>
> We thank the reviewer for the helpful suggestions. For the concerns raised by the reviewer, we respond as follows:
>
> 1. Some intuitive explanation of notations: The motivation for G_up (not G_op, as the reviewer mistakenly copied) is to define a function class that covers all optimistic policies. The matrix A in the definition of G_up is part of the confidence interval, similar to the role of the sample covariance in the construction of confidence intervals for linear contextual bandit. The motivation of Lambda_{h,t} is the sample covariance matrix which will be used to construct confidence intervals.
>
> 2. It's not clear that the LSVI-UCB algorithm is optimistic: We would argue that the optimistic nature of the LSVI-UCB algorithm is very clear, from line 10 of the algorithm that clearly appends a confidence interval term to the generalized linear estimates of the Q function. This kind of optimism term appears in many other settings, including (generalized) linear bandits, so there's no need to look into further details of the algorithm to see the optimistic structure.
>
> 3. Assumption 2 is fairly strong and not realistic: This is true to some extent, but several points are worth emphasizing. First, Assumption 2 is strictly _weaker_ than the linear MDP assumption that has become quite popular in the theoretical analysis of RL (as we show). Second, it is unlikely that these kinds of optimistic algorithms provably succeed under much weaker assumptions, indeed very recent work shows that just assuming realizability of Q^\star would be insufficient. Third, we do not know of any weaker assumptions that permit computationally and statistically tractable RL (to date, there is no computationally efficient method for the the low IBE setting of Zanette et al.). Thus our results represent the weakest tractable assumptions to-date and are close to what is information-theoretically possible.
>
> 4. Bias-variance tradeoff: While our analysis indeed assumes there exists a perfect fit of the Q functions in generalized linear forms, we would like to clarify that reporting lower regret is NOT the focus or objective of this paper. The main objective/message of this paper is to show that the regret analysis and algorithms that are previously developed for purely linear Q approximation functions can be extended to the much more general model classes discussed in this paper, thereby making the analysis/algorithm more applicable to reinforcement learning questions.
>
> We would also like to point out that Zanette et al. appeared on arxiv several months after this paper first appeared. Indeed Zanette et al., cites this paper!
>
> Minor issues: Yes we can update both the H-dependence and the venues in the bibliography

---

> > ### Comment · AnonReviewer5 · 2020-11-24
> > **I thank the authors for their clarifying remarks**
> >
> > I thank the authors for their response, it helped in understandin the author's submission. Still, after considering the author’s response, I feel my major concerns are still valid and I see little reason to change my overall assessment
> >
> > On points 1) and 2), I brought up the issues to give the authors a chance to make their submission easier to understand for other readers. For example, if the only way to understand that their approach is optimistic is by carefully examining line 10 of the algorithm then I believe there’s room for improvement in understandability. The authors seem to believe the paper is fine as is, and as understandability is very subjective we’ll have to agree to disagree on these points.
> >
> > On the weakness of assumption 2, the authors respond that “ we do not know of any weaker assumptions that permit computationally and statistically tractable RL.” This ends-justify-the-means argument does not advance science. Just because one wishes to prove something, and hasn’t come up with realistic assumptions by which to prove the thing doesn’t justify making unrealistic assumptions.
> >
> > It therefore saddens me that the authors didn’t comment on whether the roadmap for weakening assumption 2 as laid out in Zanette would work for the author’s algorithm. While I agree that Zanette’s algorithm isn’t computationally tractable while the author’s proposed algorithm is, the analysis methods used there (splitting the error into an approximation and variance error term) could, in theory, be applied to the author’s optimistic algorithm. And that is exactly what I wrote in my comment, saying that the authors could perhaps better analyze their algorithm by using the strengths of Zanette’s theoretical approach. I would have hoped the authors address this question instead of addressing the unrelated issue of the intractability of Zanette’s algorithm.
> >
> > Finally, as a comment not on the paper itself but the author's response, I find the comments about what appeared when on arxiv unbecoming for three reasons:
> >
> > - I can’t verify the authors claim without also discovering the author’s identity
> > - By claiming that “Zanette et al., cites this paper” the authors are giving hints about their true identity, which is unprofessional
> > - The intent of mentioning Zanette’s work was not to say the author’s work is unoriginal or isn’t novel, but as a hint on how the author’s can strengthen their work. Therefore, who published first is irrelevant in this context. I wish the authors had constructively commented on this instead of bringing up irrelevant who-published-first debates.

---

> > > ### Author Response · Authors · 2020-11-24
> > > **Regarding Zanette et al., and bias-variance decomposition**
> > >
> > > We highlighted the computational efficiency point above, because with current techniques it does not seem possible to get the bias-variance decomposition in a computationally tractable manner. It is not just an issue of the analysis, but rather the algorithm itself. Put another way, we do not think that the optimistic algorithm will be robust to approximation errors, due to subtleties regarding error propagation. On the other hand, we do believe that we can get a computationally _inefficient_ algorithm that is robust even in the GLM setting (which would generalize Zanette et al.), but it would look very different from our optimistic algorithm here.
> > >
> > > The point is that these two issues are _not_ unrelated at least given current techniques. We must give up robustness to approximation error to enjoy computational efficiency, so you have to choose which you care about more. Perhaps this is a matter of taste, but we felt that computational efficiency is more important than handling approximation errors. We apologize if we did not make this clear in our earlier comment.

---

### Decision · Program_Chairs · 2021-01-07
**Final Decision**

**Decision:**

Accept (Poster)

**Comment:**

This paper analyzes a version of optimistic value iteration with generalized linear function approximation.  Under an optimistic closure assumption,  the algorithm is shown to enjoy sublinear regret.  The paper also studies error propagation through backups that do not require closed-form characterization of dynamics and reward functions.

Overall, this is a solid contribution and the consensus is to accept.